# Cannabidiol-Loaded Extracellular Vesicles from Human Umbilical Cord Mesenchymal Stem Cells Alleviate Paclitaxel-Induced Peripheral Neuropathy

**DOI:** 10.3390/pharmaceutics15020554

**Published:** 2023-02-07

**Authors:** Anil Kumar Kalvala, Arvind Bagde, Peggy Arthur, Tanmay Kulkarni, Santanu Bhattacharya, Sunil Surapaneni, Nil Kumar Patel, Ramesh Nimma, Aragaw Gebeyehu, Nagavendra Kommineni, David G. Meckes, Jr., Li Sun, Bipika Banjara, Keb Mosley-Kellum, Thanh Cong Dinh, Mandip Singh

**Affiliations:** 1Department of Pharmaceutics, College of Pharmacy and Pharmaceutical Sciences, Florida A&M University, Tallahassee, FL 32301, USA; 2Department of Biochemistry and Molecular Biology, Mayo College of Medicine and Science, Jacksonville, FL 32224, USA; 3Department of Physiology and Biomedical Engineering, Mayo College of Medicine and Science, Jacksonville, FL 32224, USA; 4Department of Biomedical Sciences, Florida State University College of Medicine, 1115 West Call Street, Tallahassee, FL 32301, USA

**Keywords:** extracellular vesicles, hUCMSCs, CBD-EVs, AMPK, atomic force microscopy, morphology, Young’s modulus

## Abstract

In cancer patients, chronic paclitaxel (PTX) treatment causes excruciating pain, limiting its use in cancer chemotherapy. The neuroprotective potential of synthetic cannabidiol (CBD) and CBD formulated in extracellular vesicles (CBD-EVs) isolated from human umbilical cord derived mesenchymal stem cells was investigated in C57BL/6J mice with PTX-induced neuropathic pain (PIPN). The particle size of EVs and CBD-EVs, surface roughness, nanomechanical properties, stability, and release studies were all investigated. To develop neuropathy in mice, PTX (8 mg/kg, i.p.) was administered every other day (four doses). In terms of decreasing mechanical and thermal hypersensitivity, CBD-EVs treatment was superior to EVs treatment or CBD treatment alone (*p* < 0.001). CBD and CBD-EVs significantly reduced mitochondrial dysfunction in dorsal root ganglions and spinal homogenates of PTX-treated animals by modulating the AMPK pathway (*p* < 0.001). Studies inhibiting the AMPK and 5HT1A receptors found that CBD did not influence the neurobehavioral or mitochondrial function of PIPN. Based on these results, we hypothesize that CBD and CBD-EVs mitigated PIPN by modulating AMPK and mitochondrial function.

## 1. Introduction

The most powerful of the taxanes, paclitaxel (PTX), is used to treat advanced or metastasized forms of breast, ovarian, pancreatic, non-small-cell lung, and Kaposi’s sarcoma [1,2,3]. Patients with advanced breast cancer have been shown to benefit more from weekly PTX treatment than from nab-PTX (PTX wrapped in the protein albumin) or ixabepilone in terms of progression-free survival [4]. However, the incidence of neuropathy from PTX is one of the major clinical challenges, which has a very high prevalence ranging from 11 to 87% in cancer patients [5]. It can lead to the development of sensory dominant neuropathy manifesting clinical outcomes like paraesthesias, dysesthesia, finger and toe numbness, decreased proprioception, and loss of sensitivity [6]. These dose-dependent symptoms might last 1–3 years or even a lifetime. However, current therapies for PTX-induced neuropathic pain are partially effective, and the underpinning mechanisms for developing PTX-induced neuropathic pain are not fully understood. Therefore, there is an imperative need to develop a therapy that does not impede anti-tumor efficacy but effectively controls PTX-induced neuropathic pain (PIPN).

Cannabidiol (CBD), a non-psychoactive component of Cannabis sativa, regulates pain and is effective in managing breast cancer, psoriasis, human epithelial carcinoma, colon cancer, inflammatory bowel disease, glaucoma, and platelet aggregation. It also acts as an antidote for psychoactive cannabinoids [7]. However, CBD has poor solubility and is susceptible to degradation via the action of light and temperature in solution form and undergoes extensive first-pass metabolism [8]. Thus, a properly formulated version of CBD can play a crucial role in enhancing its physiochemical stability and therapeutic efficacy. Of note, inhalational formulations of these compounds produced irritation to the lungs, transdermal patches irritated mucosa, and nano formulations in different polymers showed entrapment and drug release problems [9].

In recent years, we have developed protocols for isolating and characterizing EVs that originate from mesenchymal stem/stromal cells (hUCMSCs). These EVs proved to be efficient in reducing neuropathic pain through their paracrine secretions. hUCMSCs-derived EVs shuttles contain biologically active substances such as proteins, lipids, mRNA, miRNA, lncRNA, circRNA, and DNA and help repair damaged nerves by stimulating the growth of new Schwann cells, activating macrophages, and rewiring blood vessels [10]. Accumulating literature also suggests that EVs derived from hUCMSCs could activate the AMPK pathway to alleviate viral myocarditis, and cannabinoids could activate hypothalamic AMPK to stimulate appetite in mouse models [11]. AMPK activation offers neuroprotection by manipulating oxidative stress, inflammation, autophagic deficits, endoplasmic reticulum stress, and mitochondrial dysfunction [12]. Therefore, the current study explored the hypothesis that CBD-EVs formulation would have an additive to synergistic effects in providing neuroprotection against PTX-induced neurotoxicity by strongly regulating the AMPK pathway.

Several lines of research have shown that EVs have lipid membranes that cover a nanosize range, making them excellent drug carriers for delivering the drug to the target areas. Further, EVs have a long half-life in the blood, cross the blood-brain barrier, and can target specific tissues without provoking an immune response [13]. EVs delivery of curcumin with improved anti-inflammatory activity compared to free curcumin was demonstrated by Sun et al. Curcumin’s solubility, stability, and bioavailability were enhanced by EV encapsulation in both in vivo and in vitro studies [14]. Schindler et al. demonstrated that exosomal delivery of doxorubicin increases in vitro cytotoxicity and speeds up cell entry [15]. Additionally, research has shown that cell-free therapeutics and gene therapy vehicles derived from hUCMSCs are promising for promoting neural regeneration [16]. This literature has demonstrated the use of EVs as potential therapeutic agents. To fully comprehend their abilities in the therapeutic regime, EVs need thorough characterization using various tools, including the atomic force microscope (AFM). AFM has become a popular tool for the characterization of various biological samples, including EVs. AFM enables label-free quantification of the sample’s nanomechanical characteristics without the need for fixation, thus, preserving their integrity [17]. Hence, studying the alteration in morphology and nanomechanical characteristics of these EVs under various treatment conditions could potentially aid in comprehending and exploiting EVs to their therapeutic capabilities. CBD has been shown to activate multiple receptors in different disease models [18]. Ward et al. demonstrated that the activation of 5HT1A receptors by CBD was responsible for improving neurobehavioral characteristics of mice against PIPN [19]. Further, Polter et al. showed that 5HT1A blocker (8-OH-DPAT) treatment reduced calcium/calmodulin dependent protein kinase II (CaMKII) phosphorylation [20]. CaMKII has the ability to activate AMPK by phosphorylation and is known to be involved in regulating mitochondrial homeostasis [21,22]. Mitochondrial dysfunction associated with dysregulation of AMPK- SIRT1 (Sirtuin 1)- NRF1/2 (nuclear respiratory factor 1/2)—TFAM (Mitochondrial transcription factor A) axis has been implicated in several neuronal diseases, which include Alzheimer’s disease, parkinsonism, diabetic neuropathy, nerve injury, Huntington’s disease, and PIPN [23,24,25]. Our western blotting findings all corroborate that CBD and CBD-EVs can modulate this pathway. We speculate that this is the mechanism through which CBD and its EVs formulation alleviate neuropathic pain by targeting mitochondrial dysfunction.

Presently, no study has been conducted to understand the role of CBD loaded in hUCMSCs-derived EVs in PIPN. In this study, we have evaluated the effect of CBD and CBD-EVs formulation on oxidative stress and mitochondrial function via targeting endocannabinoid and non-endocannabinoid receptors and the AMPK pathway. To accomplish these objectives, we have isolated and characterized the EVs from hUCMSCs by employing various techniques, including a high-resolution AFM.

Further, we investigated the pharmacological effects of CBD and CBD-EVs administration on pathophysiological indices of neuropathy in PTX-treated mice and cultured primary dorsal root ganglions (DRG) neurons (isolated from rat spinal region (L1–L5). Several studies have shown that CBD protects neurons by acting on non-cannabinoid receptors (5HT1A) rather than cannabinoid receptors (CB1 or CB2 receptors) [19,26,27]. Research has also shown that 5-HT1A agonists have strong effects against neuropathy [28]. In fact, an intraperiaqueductal grey injection of CBD causes antinociception that depends on the dose and can be stopped by giving the 5-HT1A antagonist WAY100635 [29]. Therefore, we used 5HT1A and CB1 blockers and Compound C (AMPK inhibitor) to validate the pharmacological mechanism of CBD in offering neuroprotection. The results from these studies will give significant leads and insights into the role of CBD and CBD-EVs in the treatment of chemotherapy-induced peripheral neuropathy.

## 2. Materials and Methods

### 2.1. Materials

CBD (GLP grade) was obtained from Purisys™ (Athens, GA, USA). Paclitaxel was purchased from LC Laboratories (Woburn, MA, USA), Dulbecco’s Modified Eagle Medium (DMEM) and DMEM/Ham’s F12 (1:1 Mixture) media were acquired from Millipore Sigma (St. Louis, MO, USA). Fetal bovine serum (FBS) was procured from Thomas scientific (Swedesborow, NJ, USA). Bovine serum albumin (BSA), sucrose, ethanol, methanol, water (HPLC grade), Triton X-100, formaldehyde, phosphate buffered saline (PBS, 1X) were procured from Sigma Aldrich (St. Louis, MO, USA). Human umbilical cord derived mesenchymal stem/stromal cells (hUCMSCs) of passage 0 to 2 were acquired from the Department of Medicine, Florida State University. PBS-vertical wheel bioreactor was purchased from PBS Biotech, Inc. (Camarillo, CA, USA). Cytodex-1 microcarrier was from VWR International (Radnor, PA, USA). Sodium bicarbonate and Penicillin/Streptomycin were procured from ThermoFisher Scientific (Waltham, MA, USA). 150 mm diameter Petri dishes are from Corning (Corning, NY, USA). EVs free FBS was used for EVs collection and acquired by ultracentrifugation under 100,000 rcf, 4 °C for 20 h. Cell Signaling Technology provided all primary and secondary antibodies utilized in our research. C57BL/6J female mice (4–5 weeks age) and Male Sprague dawley rats (7–8 weeks age) were obtained from Envigo (Indianapolis, IN, USA). Electric Von-frey anesthesiometer, Randall selitto paw pressure test meter and Von-frey aesthesiometer were obtained from IITC life sciences (Woodland Hills, Los Angeles, CA, USA). 

### 2.2. Methods

#### 2.2.1. Culture of hUCMSCs in PBS-Vertical Wheel (PBS-VW) Bioreactors

Dr. David Meckes laboratory at Florida State University College of Medicine supplied the hUCMSCs from passages 0 to 2. As discussed in our prior work, hUCMSCs were cultured in PBS-vertical wheel (PBS-VW) bioreactors to get optimal results when isolating EVs [30]. Complete culture media for human umbilical cord derived mesenchymal stem cells (hUCMSCs) was: α-MEM supplemented with 10% EV free FBS, sodium bicarbonate, and 1% Penicillin/Streptomycin. For bioreactor studies, cells were detached using Trypsin-EDTA at 80–90% confluence. We used a 0.1L PBS-VW bioreactor and seeded 1100–1500 cells/cm^2^ with 0.25 g of cytodex-1 as micro carriers. 60 mL of media was utilized for the initial stage of seeding the cells, and the speed was adjusted at 25 rpm for five minutes, followed by a stationary state for 15 min, throughout the course of four hours. Isolating EVs from cell-conditioned media was accomplished by using a modified differential centrifugation approach that included polyethylene glycol precipitation. Size and zeta potential of EVs were determined with the aid of ZetaView^®^ BASIC NTA—Nanoparticle Tracking Video Microscope PMX-120, ZetaView software (version 8.05.11 SP4, Particle Metrix, Meerbusch, Germany). The expression of EVs-recommended gold standard markers CD81, CD 63, alix, and flotillin-2 was measured by Western blotting [31,32].

#### 2.2.2. Atomic Force Microscopy

EVs were analyzed by AFM post-adsorption onto a mica sheet that was modified using 3-aminopropyltriethoxysilane (APTES) [33]. A fresh layer of muscovite mica sheet was glued onto a Teflon sheet and then onto an AFM metal specimen disc. After slicing off the mica’s top layer, it was treated with a mixture of APTES and N,N-diisopropylethylamine (DIPEA) for 2 h at 60 °C before being used. The original EV stock solution was diluted one thousand times with Milli Q water solution to create a working sample of EVs. The APTES-modified mica was drop-casted with 5 µL of the working solution. To get rid of any loose EVs, the mica was rinsed in distilled water after 30 min. Afterward, we employed a Dimension Icon Scanasyst AFM (Bruker Corporation, Santa Barbara, CA, USA) on the sample and applied peak-force quantitative nanomechanical mapping to study its behavior in a fluid environment (PFQNM). To obtain such fine detail in the EVs topography, a Scanasyst Air probe with a pyramidal tip was used. This sharp-tipped probe has a 5 nm radius. The probe was not optimized for characterizing topographical morphology. The topography was evaluated at a scan rate of 0.1 Hz and a peak force of 300 pN. The EVs’ morphological features, including their height and surface roughness, were also analyzed using Nanoscope Analysis v1.9 software (Nanoscope Technologies, Bedford, TX, USA). Root-mean-squared variation in feature height was used to calculate the surface roughness. The spring constant of 0.38 N/m and the deflection sensitivity of 35 nm/V was determined by calibrating the AFM probe in a fluid environment on a plain mica surface for the purpose of characterizing nanomechanical characteristics. At least 50 samples were subjected to a nanoindentation experiment, with each tip-sample interaction yielding a force-separation (F-S) curve. In addition, the Young’s modulus (YM), deformation, and adhesion were extracted from the F-S curves using the DMT model.

#### 2.2.3. Preparation of Optimized CBD-Loaded Extracellular Vesicles and Characterization

An optimized method for the preparation of CBD-EVs developed in our lab was used [30]. In brief, CBD was loaded into EVs using a previously standardized sonication method. Blank EVs (1.5 × 1011 particles/mL) were incubated with 10% *w*/*w* (in terms of protein content) CBD solution and subjected to different sonication cycles (20% Amplitude, 3 cycles of 30 s on/off for 2 min, 5 min-cooling between each cycle). To obtain a stable formulation with no precipitation, EVs were stabilized with 0.1% (*w*/*v*) BSA and a 10% sucrose solution. By using nanoparticle tracking analysis, the formulation’s physical appearance, size, particle number, and zeta potential were evaluated [19]. The entrapment efficiency was assessed by ultrafiltration method using Vivaspin^®^ 500 centrifugal filter unit (Sartorius, Bohemia, NY, USA) and RP-HPLC analysis as reported earlier [34].

#### 2.2.4. Release Studies

The modified dialysis bag method was used to investigate the in vitro drug release as per the previously described method elsewhere [35,36,37]. CBD was loaded into EVs and enclosed in a dialysis membrane pouch (12 kDa). This pouch was placed in 10 mL of PBS with 0.5% tween 80 at pH 6.8 and 7.4. The complete apparatus was kept in a 120-rpm, 37 °C shaker bath. At intervals, 1 mL of the sample was taken and replaced with fresh medium to create sink conditions. CBD content was measured using the RP-HPLC method.

#### 2.2.5. Animals

C57BL/6J female mice (4–5 weeks age) and Male Sprague dawley rats (7–8 weeks age) were obtained from Envigo (Indianapolis, IN, USA) and were used to conduct peripheral neuropathy studies. Florida Agricultural and Mechanical University has AAALAC-accredited animal facilities, and all the animal experiments carried out were reviewed and approved by the Institutional Animal Use and Care Committee of Florida Agricultural and Mechanical University (protocol numbers: 020-06 and 021-04) in compliance with NIH guidelines (Guide for the care and use of laboratory animals). All mice and rats were euthanized via exposure to carbon dioxide (CO_2_). 

Briefly, animals were divided into following groups (n = 6/group): (a). Normal Control: Age-matched untreated mice, (b). Paclitaxel (PTX) group: 8 mg/kg of PTX administered (i.p.) every other day for four injections), (c). PTX+EVs: Mice were given extracellular vesicles equivalent to load CBD (5 mg/kg) body weights of mice (i.p) twice a week for total of six weeks after the last dose of PTX injection. (d). Paclitaxel (PTX) + Cannabidiol (CBD): 8 mg/kg of PTX administered (i.p.) every other day for four injections and 5 mg/kg of CBD (i.p) twice a week for a total six of weeks after the last dose of PTX injection, (e). PTX + CBD + EVs: 5 mg/kg of CBD and hUCMSCs-derived EVs (i.p.) injected twice a week for six weeks after the last dose of PTX injection and f. Paclitaxel (PTX)+Cannabidiol EVs (CBD-EVs): 8 mg/kg of PTX administered (i.p.) every other day for four injections and 5 mg/kg of CBD loaded in hUCMSCs derived EVs (i.p) twice a week for a total of six weeks after the last dose of PTX injection, PTX + WAY + CBD: 10 mg/kg/day, i.p., WAY100135 (5HT1A receptor inhibitor) was given to mice for four weeks and three hours before administering CBD, PTX + CBD + RIMO: 3 mg/kg/day, i.p., rimonabant (CB1 receptor blocker) was given to mice for four weeks and three hours before administering CBD. After confirming neuropathy in the mice, the neurobehavioral changes of the animals in the different groups were assessed twice a week for a total of 42 days. At the end of the experiments, the animals were sacrificed, and spinal cords and dorsal root ganglions were isolated from L1 to L5 region of the spinal cord, followed by biochemical and molecular assessments. The experimental design and dose of CBD was selected based on the previous reports [19,38].

#### 2.2.6. DRG Primary Cultures

All neuronal cultures were prepared from (200–250 g) Sprague dawley rats aging 7–8 weeks, and DRGs were dissociated using previously reported methods with slight modifications [39]. Under sterile conditions, DRGs with roots ((L1–L5); 25–30/Rat) were collected in F-12 (Hams-F 12) medium supplemented with 10% Fetal bovine serum, and ganglia were carefully removed from roots and capsular connective tissue. The tissue was incubated with collagenase (0.125%) for 1 1/2 h followed by centrifugation (1200 RPM, 2 min) and then trypsin (0.25%) incubation for 30 min and triturated with a glass pipette to dissociate into cells. Trypsin was deactivated by adding 20% FBS, and then the suspension was filtered through a 70 micrometers (μm) nylon gauge and centrifuged at 1200 RPM for 5 min; the supernatant was removed, and the cell pellet was resuspended in primary neuron basal media (cat#CC-3256, LonZa, Rockville, MD, USA) containing primary neurons supplement (cat#CC-4462, LonZa, Rockville, MD, USA). Single rat DRGs isolated were used to make two 24 well plates with 10,000 cells/well. The 24 well plates were coated with 50 μL of matrigel and wrapped over the bottom of the plate, and dried for 1 h in a laminar hood before plating the cells.

#### 2.2.7. Behavioral Parameters

##### Thermal and Mechanical Hyperalgesia

Thermal hyperalgesia in mice was measured by using Hargreaves plantar test and tail immersion method as described earlier [40,41].

a.Plantar Test (Hargreaves Method)

Prior to the experiment, the animals were acclimatized for 1 h on a heated base (30 °C) horizontal glass surface covered with plexiglas boxes which can accommodate 12 mice at a given time. Using test head and radiant heat source, the time taken for a mouse to lift the paw when infrared irradiation (40 IR units) was exposed was recorded with a cut-off time of 20 s, and five consecutive readings were recorded by giving a 10 min time gap, and each reading reported as paw withdrawal latency in seconds.

b.Hot immersion test

The sensitivity of the mice to heat was evaluated with a hot (55 °C ± 0.5) test, in which the number of seconds it took for the animal to flick its tail was recorded; three separate readings were taken at 10-min intervals to account for inter-reading variability, and the average value was reported as paw withdrawal latency in secs.

c.Electronic Vonfrey Test and Randall Selitto Test

Mice paw withdrawal threshold is measured electronically using a von Frey gram scale. There was at least a 10-min gap between each of the five readings. We took the mean of each animal’s five observations to determine its paw withdrawal threshold. Randall selitto pincture pressure was applied on both paws, and paw withdrawal time point was recorded with a difference of 10 min in the recording between two consecutive readings. The average of five readings per animal was reported as paw withdrawal pressure in seconds [42,43].

#### 2.2.8. Biochemical and Molecular Parameters 

##### Estimation of ATP Levels

The ATP content of homogenized DRG tissues was calculated using the manufacturer’s kit guidelines (calorimetric assay, MAK190, Sigma, St. Louis, MO, USA). As has been previously reported, the ATP concentration was given in terms of nmol/mg protein [40].

##### Estimation of NAD^+^ Levels and NADH Levels

NAD^+^/NADH levels in the fresh DRG homogenates were measured according to the manufacturer’s protocol (#MAK037-1KT, Sigma Aldrich, St. Louis, MO, USA). The concentration of NAD^+^ levels was expressed in nanograms/mg protein as described earlier [3,42].

##### JC1 Assay

JC-1Staining was performed on dissociated fresh DRG cells as described earlier [23,39]. For 30 min, 5 µM JC-1 stain was added to dissociated DRG cells. After centrifuging at 3000 rpm for 5 min at 37 °C, the pellet was resuspended in PBS. After adding 200 uL of JC-1 treated cell suspension in PBS to each black well, we monitored the red fluorescence emitted by the cells using an Infinite M200 multi-plate reader at a wavelength of 590 nm (Tecan, Grödig, Austria). The red fluorescence intensity was adjusted with protein content and expressed as a percentage compared to the control group.

##### Western Blotting

DRG and spinal cord tissue protein homogenates were prepared in tissue protein extraction reagent (TPER, Sigma, St. Louis, MO, USA), and DRG primary cell lysates and hUCMSCs and hUCMSCs-EVs lysates were prepared in radioimmunoassay buffer (RIPA) containing 1:100 protease (#P1860, Sigma Aldrich, St. Louis, MO, USA) and phosphatase (#P0001, Sigma Aldrich, St. Louis, MO, USA) inhibitors. The tissue and cellular homogenates were centrifuged at 10,000× *g* for 20 min at 4 °C, and the supernatants were collected, and the protein content was determined using a bicinchoninic acid assay kit. Briefly, 40 µg of protein samples were loaded and resolved by using SDS-PAGE gel electrophoresis and transferred onto the PVDF membrane using Transblot “Turbo” transfer system (semi-dry transfer unit, BIORAD, Hercules, CA, USA) followed by blocking with 5% BSA solution in PBST. After blocking, The PVDF membranes were incubated with primary antibodies at 4 °C overnight; AMPK, p-AMPK (Thr 172), SIRT1, NRF2, NRF1, SOD2, SIRT3, 5HT1A, and Parkin (Cell signaling technology, Danvers, MA, USA), Flotillin-2, CD 63, CD81, Alix, Calnexin, Catalase, TFAM, HO-1, CB1 and NQO1 (Santacruz biotechnologies, Santa Cruz, CA, USA) were prepared at 1:1000 dilution in PBST. The membranes were incubated with HRP-conjugated secondary anti-rabbit and anti-mouse antibodies for 2 h at room temperature. The luminescence signal was captured using a ChemiDoc^TM^XRS^+^ imaging system (BIO-RAD), and the obtained band intensities were quantified by using image J software (version 1.48, NIH, Bethesda, MD, USA) [44,45,46].

##### Neurite Outgrowth Assay

Briefly, 10,000 DRG cells were seeded in each well of a 24-well plate and then subjected to 3 µM of PTX for 48 h before being treated with CBD and CBD-EVs at 12 µM each. At random, five fields were chosen to be analyzed with a phase contrast microscope (Nikon ECLIPSE, Ti-U, Tokyo, Japan). Using Image J (National Institutes of Health, Bethesda, MD, USA), we measured the length of neurite outgrowths in 30 cells from each experimental area. Neurite outgrowths/axon-like extensions with a diameter greater than the cell body were counted [47].

##### Immunocytochemistry

Dissociated rat DRG primary cells were seeded at 5000 cells/well in 6-well plates. After 48 h, the cells were taken from the culture, washed three times with PBST, and fixed in 4% paraformaldehyde for 15 min. Before blocking with 3% BSA for 2 h, cells were permeabilized with 0.5% Triton-X 100. DRG primary cells were treated overnight with anti-rabbit p-AMPK (Thr 172) in 3% BSA. The next day, cells were washed with PBST and treated with anti-mouse rhodamine for 2 h at room temperature. Coverslips were placed using DAPI mounting media (FluoroshieldTM, Sigma, St. Louis, MO, USA). Confocal images were taken (Leica TCS SP8 Laser Scanning Spectral Confocal microscope, Wetzlar, Germany) [23,37].

#### 2.2.9. Statistical Analysis

To compare the groups, one-way analysis of variation (ANOVA) was used, followed by post hoc analysis using Bonferroni’s Multiple Comparison Test in GraphPad prism software (9.5.0 (730), GraphPad Software, San Diego, CA, USA). To compare more than two factors, two-way ANOVA was used, followed by Bonferroni’s Multiple Comparison Post Test. All data were analyzed in consultation with a statistician and was considered statistically significant at *p* < 0.05.

## 3. Results

### 3.1. Isolation and Characterization of Human Umbilical Cord Mesenchymal Stem Cell-Derived Extra Cellular Vesicles (EVs)

The isolation of EVs from media collected from hUCMSCs cultures grown in the 0.1 L PBS-VW bioreactor using a modified differential centrifugation method with PEG precipitation was successful, as reported previously [30]. Cytodex 1 microcarrier was used to facilitate the growth of the hUCMSCs in the bioreactor by acting as a support matrix. The mean particle size, average particle number, and zeta potential of the isolated EVs were 141.4 ± 2.1 nm, 3.6 × 10^10^ particles/mL, and −21.29 ± 0.66 mV, respectively. The average protein concentration obtained by using the BCA assay kit was approximately 1.1 mg/mL.

Consequently, isolated total proteins from hUCMSCs lysates and EVs homogenates are resolved in SDS PAGE gel electrophoresis. Tetraspanins like CD 63 and CD 81, which are considered exosomal markers, increased two to four folds (*p* < 0.001) in protein lysates obtained from EVs when compared to hUCMSCs cell lysates (Figure 1a,b). Other exosomal markers, flotillin 2 and alix, were also significantly (*p* < 0.001) increased in EVs lysates when compared to hUCMSCs lysates, as shown in Figure 1a,b. Further, we also probed for exosomal negative marker calnexin, which significantly (*p* < 0.05) increased in hUCMSCs lysates when compared with EVs protein lysates (Figure 1a,b). The CBD was loaded into EVs via the optimized sonication method consisting of bovine serum albumin (BSA) pH 7.4, 10% *w*/*w* sucrose, with sonication conditions (3 cycles of 30 s on/off, 20% Amplitude (Amp), for a total of 2 min out of 5 min-cooling periods between each cycle), followed by incubation at 22/37 °C for 1 h. The mean particle size of EVs loaded with 10% CBD after sonication was 131.9 ± 1.1nm (Figure 1c) and entrapment efficiency of 92.3 ± 2.21%, and a zeta potential of −30.26 ± 0.12 mV.

### 3.2. Morphology and Nanomechanical Attributes Characterization of Human Umbilical Cord Mesenchymal Stem Cell-Derived EVs for Various Treatments

Mica surface has been commonly employed to study both organic and inorganic samples as mica provides an optimally flat surface with a minimal negative surface charge [48].

The flatness of the mica surface was determined using the AFM from both the height image as well as peak force error image, as shown in Figure 2a,b, respectively. Since EVs possessed a negative charge, the surface of plain mica was modified with 3:1 APTES: DIPEA to make its overall surface charge positive for effective EVs adsorption. Such treatment did not introduce any significant topographical features, as seen from the corresponding height and peak force error images, in Figure 2c,d, respectively. Figure 2e,f display height and peak force error images, respectively, for human umbilical cord mesenchymal stem cells (hUCMSC)-derived EVs. As a control, we also performed morphology characterization on hUCMSC-derived EVs subjected to sonication, as shown in Figure 2g,h. Lastly, height and peak force error images corresponding to hUCMSC-derived EVs loaded with CBD drug is shown in Figure 2i,j, respectively. We used these EVs combinations to further quantify their height and surface roughness, as shown in Figure 2k,l, respectively. We did not observe any significant alteration in the height of hUCMSC and sonicated hUCMSC-derived EVs bearing values corresponding to 38.63 ± 3.32 nm and 39.83 ± 4.65 nm, respectively. However, we observed a significant increase in the height value of CBD EVs with a value of 55.03 ± 24.42 nm, as seen in Figure 2k. A similar trend in average surface roughness was observed for the EVs as seen from Figure 2l. While hUCMSC and sonicated hUCMSC-derived EVs with average surface roughness values corresponding to 2.75 ± 0.43 nm and 2.87 ± 0.36 nm exhibited no significant alterations, CBD EVs displayed significantly higher average surface roughness corresponding to 4.9 ± 0.37 nm as seen from Figure 1l.

We then performed mechanical attributes characterization of hUCMSC-derived EVs under various modifications by analyzing each corresponding F-S curve. Figure 3a shows a representative F-S curve displaying an overlap between the approach (tip approaching the sample) and retract (tip retracting from the sample), which indicates a complete tip-sample retraction. The retrace part of the curve was then used to calculate Y_M_ by fitting the curve using the DMT model. Figure 3b,c display F-S curves corresponding to sonicated and CBD EVs. We observed a slight decrease in the slope of the retrace curve for sonicated hUCMSC-derived EVs compared to hUCMSC-derived EVs with no treatment. This was confirmed in Figure 3d, where the Y_M_ for sonicated hUCMSC-derived EVs was 6.11 ± 0.67 MPa and higher than hUCMSC-derived EVs with no treatment bearing average Y_M_ value of 5.57 ± 0.45 MPa. However, the slope of the retrace curve for CBD-loaded hUCMSC-derived EVs was maximum and confirmed from the corresponding Y_M_ value of 8.66 ± 0.57 MPa, as seen in Figure 3d. The average deformation values corresponding to hUCMSC-derived EVs with no treatment and sonicated hUCMSC-derived EVs showed no significant alteration corresponding to the values 20.12 ± 2.8 nm and 21.86 ± 4.18 nm, respectively. However, the average deformation for CBD EVs was observed to be 13.28 ± 2.12 nm, as seen in Figure 3e. Lastly, we quantified the adhesion parameter, which quantifies the repulsive pull the tip experiences from the sample upon the cusp of retraction indicated by the sharp dip in force value before the tip retraction, as shown using dotted line in F-S curves in Figure 3a–c. Upon quantification, the average adhesion was found to be maximum in CBD EVs with a value corresponding to 115.5 ± 5.76 pN. No significant change was observed between sonicated hUCMSC-derived EVs and hUCMSC-derived EVs with no treatment, with values corresponding to 81.41 ± 5.52 pN and 75.85 ± 7.97 pN, respectively. However, these values were significantly lower than CBD-loaded hUCMSC-derived EVs. These distinguishing morphologies and nanomechanical attributes corresponding to hUCMSC-derived EVs using AFM provide a new avenue to commonly explored characteristics such as particle size, zeta potential, and stability.

### 3.3. In Vitro Drug Release from CBD-Loaded EVs

The in vitro drug release experiments for CBD-loaded EVs (100 μg/mL) was performed by dialysis method with cellulose acetate dialysis tubing at pH 6.8 and 7.4 in PBS. At both pH conditions, CBD depicted a sustained release in 24 h. From Figure 1d, the percent cumulative release of CBD after 24 h at pH 6.8 and 7.4 were 50.74 ± 2.44% and 53.99 ± 5.04%, respectively.

### 3.4. Effect of CBD and CBD-Loaded EVs on Neurobehavior of PTX-Treated Mice

Following PTX administration, tail and paw withdrawal latencies to thermal stimuli were significantly reduced after six weeks in hot water immersion/hot plate method (*p* < 0.001, 4.05 ± 0.31 vs. 10.86 ± 0.36), cold plate method (*p* < 0.001, 5.78 ± 0.24 vs. 13.43 ± 0.43), and hargreaves plantar test (*p* < 0.001, 3.85 ± 0.8 vs. 10.89 ± 0.40). The decreases in tail flick and paw withdrawal latencies with hot/cold stimulus and IR radiation were reversed when treated with 5 mg/kg CBD, EVs, or CBD-EVs (Figure 4a,b/Table 1). Further, PTX-treated mice had lower paw withdrawal thresholds (1.91 ± 0.36 g vs. 4.63 ± 0.25 g) (*p* < 0.001) and pressures (84.47 ± 8.72 g vs. 171.05 ± 5.69 g) (*p* < 0.001) than age-matched control mice (Figure 4c,d/Table 1). CBD (*p* < 0.05), EVs (*p* < 0.05), and CBD-EVs (*p* < 0.01) at 5 mg/kg corrected these mechanical neurobehavioral changes (Figure 4c,d/Table 1). The results show that CBD-loaded EVs outperformed CBD and EVs alone in correcting the animals’ neurobehavior after PTX administration. A 5HT1A blocker (10 mg/kg i.p.) reduced mechanical and thermal hypersensitivity, but not rimonabant (3 mg/kg i.p.), which had no effect on the neurobehavior of CBD-treated animals (Figure 4).

### 3.5. Effect of CBD and CBD-Loaded EVs on AMPK Pathway

AMPK is a bioenergetic sensor that regulates bioenergetic pathways that aid in the maintenance of cellular homeostasis. When compared to an age-matched normal control group, p-AMPK (Thr 172) was significantly (*p* < 0.001) reduced in DRG and spinal homogenates of PTX-induced neuropathic mice (Figure 5). Furthermore, p-AMPK (Thr 172) positively regulated proteins SIRT1, GABP Alpha, TFAM, SOD2, NQO1, PINK1, and HO1 protein expression was significantly (*p* < 0.001) decreased, and negatively regulated NF-kB protein expression was significantly increased in DRG homogenates of PTX-treated mice compared to age-matched normal control group (Figure 5). Interestingly, as shown in Figure 5, these protein expressions recovered after treatment with CBD and CBD-EVS at 5 mg/kg doses.

Additionally, in PTX-treated mice, spinal homogenates showed significantly lower protein expressions of SOD2 (*p* < 0.001), Parkin (*p* < 0.001), SIRT1 (*p* < 0.001), SIRT3 (*p* < 0.05), and catalase (*p* < 0.05). CBD and CBD-EVs treatment significantly increased the expression of these proteins, as shown in Figure 6a,b. In PTX-induced neuropathic mice, CBD-EVs treatment was found to be more effective than CBD alone in normalizing these protein expressions. CBD also significantly increased the expression of the 5HT1A receptor in PTX-treated mice DRG homogenates (Figure 6c,d).

### 3.6. Effect of CBD and CBD-Loaded EVs on Mitochondrial Function in DRG and Spinal Homogenates of PTX-Treated Mice

ATP and NAD^+^ levels were significantly (*p* < 0.001) reduced in isolated fresh DRG and spinal homogenates of PTX-treated neuropathic mice (Figure 7). However, CBD-EVs treatment significantly (*p* < 0.001) increased these levels in DRG and spinal homogenates of PTX-treated mice (Figure 7). Unlike, CBD treatment showed a significant (*p* < 0.01) increase of NAD^+^ levels in DRG homogenates (Figure 7b) and ATP levels in spinal homogenates (Figure 7d) of PTX-treated mice. Further, fluorescence assay carried out in dissociated DRG primary cells isolated from mice (L1–L5 region) using mitoprobe JC1 assay kit demonstrated mitochondrial membrane repolarization effects with CBD and CBD-EVS treatment as shown by the concentration-dependent formation of red fluorescent J1 aggregates in DRG mitochondria (PTX-treated mice, Figure 7c). This data suggests that CBD-EVs have a superior mitoprotective effects than CBD treatment alone.

### 3.7. AMPK Dependent Neuroprotective Effects of CBD in PTX-Treated Primary Rat DRG Neurons

We also studied the neuroprotective effects of CBD in the presence of compound C (dorsomorphin dihydrochloride, Sigma, St. Louis, MO, USA), a potent AMPK inhibitor (K_i_ = 109) to confirm the CBD-dependent effects on AMPK pathway in DRG primary cultures. Based on the literature, a concentration of 10 µM of compound C was chosen as the AMPK inhibitory concentration [49,50]. The number of neurite outgrowths/axon-like extensions, which were double/more than the diameter of the cell body, was noted. The neurite outgrowths and percentage of neurite-bearing cells were significantly (*p* < 0.001) reduced in PTX-treated DRG cells when compared to untreated DRG cells (Appendix A). 

CBD and CBD-EVs treatment significantly (*p* < 0.001) improved the neurite outgrowth and percentage of neurite bearing-cells when compared to PTX-treated primary DRG cells. Interestingly, CBD treatment failed to improve the neurite outgrowths in PTX-treated primary DRG cells after treatment with compound C (Appendix A). Moreover, Immunocytochemistry analysis in DRG Primary cultures also revealed reduced expressions of p-AMPK (Thr 172) when treated with PTX at 3 µM for 48 h (Figure 8a). CBD and CBD-EVs treatment at 12 µM increased the expression of p-AMPK (Thr 172) in PTX insulted primary DRG neurons (Figure 8a). 

However, CBD failed to increase this protein expression in the presence of 10 µM compound C in PTX insulted primary DRG neurons (Figure 8a). Further, we explored the downstream signaling of AMPK in the presence of compound C by performing western blotting. The expressions of p-AMPK (Thr 172), SirT1, HO-1, NRF1, TFAM, and Catalase were not altered in PTX + CBD + CC treated group when compared to the PTX group, but these proteins were significantly increased in CBD and CBD-EVS treated primary DRG homogenates (Figure 8c,d). These results suggest that the neuroprotective and mitoprotective effects of CBD possibly depend upon the activation of AMPK.

## 4. Discussion

PTX is a commonly used anticancer drug for treating lung, breast, prostate, and gynecologic cancers. However, PTX causes irreversible neuropathy, limiting its clinical use in cancer chemotherapy. Neurotropic factors secreted by hUCMSCs promote neurite outgrowth and repair damaged peripheral neurons [51]. Schwann cell exosomes improved DRG cell proliferation and slowed apoptosis in injured DRG cells [52]. However, no research has been done on the role of hUCMSC exosomes/EVs in regulating neuropathic pain. This would be the first study to show that CBD-encapsulated EVs protect against PIPN.

The lack of stability of CBD as a therapeutic agent is hindered by its oxidation. Our lab developed CBD-EV formulations to overcome this limitation and recently reported their importance in treating triple-negative breast cancer in athymic nude mice [30]. Characterization of EVs has relevance for understanding their underlying contributions to tissue homeostasis or disease pathophysiology, as well as for diagnostic or therapeutic applications. Numerous approaches are available for the morphological characterization of EVs, such as electron microscope (EM), Raman spectroscopy, resistive pulse sensing, fluorescence correlation spectroscopy, stimulated emission depletion microscopy, or enzyme-linked immunosorbent assays. Several of these approaches, such as transmission EM or cryo-EM, can provide detailed morphological characterization at the level of single EVs [53,54]. However, AFM allows for 3-dimensional morphology characterization unlike other abovementioned techniques, which is evident from the height image and its quantification, as shown in Figure 2k. In addition, surface roughness characteristics convey the topographical variations at the surface of these EVs due to various treatments, as shown in Figure 2l. Recently, one of the studies focussed on streamlining the nanomechanical and morphology characterization of EVs to overcome the laborious approach of acquiring these attributes from a single EV. They focused on EVs derived from three different sources viz. human colorectal carcinoma cell culture, raw bovine milk, and Ascaris suum nematode excretions. Using this technique, the authors were able to segregate between the subpopulations of vesicular and nonvesicular objects as well as between populations of vesicles with similar sizes but dissimilar nanomechanical characteristics [55]. Another study focused on the advantages of the AFM tool for screening EV populations derived from the human neural stem cell line (hNSC) CTX0E03. and identify the trend correlating with biochemical alterations. The authors demonstrated the capability of the AFM tool to identify subtle differences in structural and mechanical properties of the derived EVs using different techniques such as ultracentrifugation and sonication. EVs prepared via ultracentrifugation technique were observed to possess slightly elevated physical dimensions and decreased adhesion coinciding with reduced CD63 levels. On the other hand, EVs prepared by sonication displayed a decrease in size and adhesive force along with reduced CD81 expression [56]. However, both of these studies do not discuss the potential therapeutic applications of EVs.

Here, combining NTA and AFM data, we observed that the overall effect of CBD-loaded drug on HUCMSC EVs reduced their overall size slightly but made them bulky indicated by the AFM height profile data. Biomechanical properties of EVs are not readily discernible by TEM and other previously mentioned methods. AFM is distinctive in allowing for the characterization of EVs in their natural fluid environment. This preserves their true shape and biomechanical properties, unlike studies performed in air medium, such as TEM, in which EVs are dried and lose their characteristic biomechanical properties [57,58]. Studies using AFM to characterize size distribution or nanomechanical properties of cell-derived EVs have been reported [59,60]. While our study is a first report on the characterization of HUCMSC-derived EVs with and without CBD-loaded drug, comparisons can be drawn with AFM studies of EVs derived from other sources [61]. AFM study conducted by Sharma et al. demonstrated the nanomechanical profile of U87 EVs using the peak force mapping (PFM) technique, which quantified their Young’s modulus and adhesion to be 2GPa and 1.6nN, respectively. Recently, living exosomes were isolated from bone marrow samples of lymphoma patients and characterized using the AFM-PFM technique, in which various nanomechanical properties such as Young’s modulus, adhesion, deformation, and energy dissipation across the EV profile were monitored [62]. PFM technique at times can provide unrealistic values due to inadequate interaction between the tip and sample surface. In such cases, it becomes imperative to consider the accuracy of F-S curve corresponding to these values, which often gets neglected. Our previous studies have shown that the nanomechanical properties of soft biological samples highly depend upon the type of probe and the samples intrinsic properties [63]. Hence, caution needs to be exercised while comparing nanomechanical attributes of EVs derived from the same/different source. In this study, specific alterations in the surface roughness, height, stiffness, deformation, and adhesion of hUCMSC-derived EVs occurred in CBD-loaded EVs compared to EVs devoid of CBD. Analyzing each F-S curve using the DMT contact mechanics model ensured accurate values corresponding to their nanomechanical properties. Such nanomechanical attributes could potentially aid in understanding the underlying mechanisms behind alleviating discomfort and pain management.

The CBD-EVs were formulated, and the release studies showed that more than 50 percent of CBD was released by 24 h at pH 6.8 and 7.4. Moreover, exosomes express definite endosomal pathway markers, including tetraspanins (CD63 and CD81), heat shock proteins (HSP70), and Rab family proteins such as TSG101 and Alix [13]. Consistent with earlier reports, we also observed the expression of tetraspanins, alix, and flotillin 2 in exosomal lysates derived from hUCMSCs. Calnexin was considered as a negative protein marker of exosomes [64] which is also under-expressed in hUCMSCs-EVs lysates.

In the present study, mechanical and thermal pain sensitivity increased in PTX-treated mice, and the reversal of this pain by CBD administration was consistent with the published reports [19]. The statistical significance in reversing the PTX-induced pain in mice with CBD administration was two folds less when compared with CBD-EVs formulation, which is an interesting observation of the current study. Lee et al. showed that EVs (derived from hUCMSCs) administration recuperated nerve ligation-induced mechanical and thermal hypersensitivities of rats [65]. Additionally, hUCMSCs itself decreased the pain hypersensitivity associated with chronic constriction injury, spinal nerve ligation, and spared nerve injury in rats [66]. These findings suggest that the addition of EVs with CBD may have additive/synergistic effects in attenuating PTX-induced neuropathic pain.

PIPN is associated with the development of spontaneous activity and hyperexcitability in DRG neurons which contributes to tremendous pain [67]. In our study, we also observed molecular changes in isolated DRGs from PTX-treated mice and PTX-treated cultured DRGs isolated from rats. PTX is known to reduce mitochondrial function and thereby precipitate sensory neuropathies. PTX treatment halted the growth of neurites in cultured DRGs, which has also been suggested in earlier reports [68]. Interestingly, CBD and CBD-EVs increased neurite outgrowths in DRGs and the length of neurites against PTX insult in cultured DRG cells. Axonal degeneration, loss of large fibers, and decrease in myelinated fiber density have all been linked to paclitaxel in clinical studies [69,70]. In this study, we looked at the effects of paclitaxel on neurite outgrowths in adult rat DRG neurons and observed that CBD and CBD-EVs formulations reversed the axonal degeneration caused by PTX. These CBD and CBD-EV-mediated axonal regenerations may contribute, at least in part, to the reversal of PTX-induced neuropathy. Further, compound C treatment prior to CBD treatment showed no improvement in PTX-induced neurite outgrowth reduction in cultured DRGs. This data suggests that CBD-EVs have superior effects in improving neural circuits against PTX-induced disturbances. However, CBD may enhance this neurite outgrowth through the activation of AMPK, as evidenced by showing its neutral effects in the presence of compound C, which is a well-known AMPK inhibitor and has been articulated in several reports [40,71]. 

In the present study, we also observed that PTX administration significantly decreased the ATP and NAD+ levels in both DRG and spinal homogenates of mice which directly correlates to unhealthy mitochondria. Prior research has shown that the cell bodies of sensory neurons undergo changes before, during, and after paclitaxel-induced pain. Prior to the onset of pain behavior, paclitaxel in vivo acutely causes impairments in mitochondrial bioenergetics in DRG neurons. DRG neurons preferentially switch to glycolysis during paclitaxel-induced pain, which may account for the prolonged fall in ATP levels [72]. CBD markedly improved mitochondrial biogenesis and mitochondrial function against doxorubicin-induced cardiotoxicity [73]. Likewise, in this study, CBD treatment partially enhanced ATP and NAD^+^ levels in both DRG and spinal homogenates of PTX-treated mice. However, we observed that CBD-EVs formulation improved ATP and NAD+ levels in a superior fashion than CBD alone in both DRG and spinal homogenates of mice. To support this data, a study conducted by Zhou et al., demonstrated that EVs derived from hUCMSCs expressed PINK1 protein and treatment with them improved mitochondrial homeostasis against sepsis-induced mitochondrial dysfunction in cardiomyocytes and also, Gu et al. reported that exosomes isolated from hUCMSCs reduced the severity of viral myocarditis by inhibiting apoptosis in cardiomyocytes via the AMPK/mTOR signaling pathway [11,74]. Moreover, we also observed reduction of AMPK and PINK1 protein expression in DRG homogenates of PTX-treated mice and significantly increased expression of AMPK and PINK1 after treatment with CBD and CBD-EVs. The enhanced improvement of ATP and NAD^+^ levels with CBD-EVs may be attributable to the expression of PINK1 in EVs derived from hUCMSCs. Moreover, EVs derived from hUCMSCs improved mitochondrial function by increasing mitochondrial membrane potentials as evidenced by JC1 staining in human pulmonary microvascular endothelial cells against LPS-induced cellular damage [75], and CBD also enhanced mitochondrial repolarisation in macrophages and microglia cells against saturated palmitic acid exposure [76]. Consistent with previous reports in this study, we also observed CBD treatment prevented PTX-induced drop in mitochondrial membrane potential in dissociated DRG cells of mice [77]. The improvement of mitochondrial membrane potentials with CBD-EVs formulation was two folds higher than with CBD alone treatment. Hence with previous reports and the current study, we can conclude that CBD-EVs formulation may have a synergistic effect in enhancing mitochondrial membrane potentials.

AMPK is considered a metabolic manipulator as well as an energy-sensing kinase and has emerged as a novel pharmacological target for the treatment of chronic pain due to its ability to regulate oxidative stress, inflammation, mitochondrial function, biogenesis, autophagy, and endoplasmic reticulum stress [12]. Our study demonstrated that PTX treatment downregulated the expression of p-AMPK (Thr 172) in both DRGs of in vitro and in vivo studies suggesting its role in PIPN. Similarly, studies conducted by Inyang et al., also reported that the administration of AMPK activators alleviated PTX-induced mechanical hypersensitivity and hyperalgesic effects in male and female mice [78]. Indeed, cannabinoids can improve AMPK activity by different signaling mechanisms in improving appetite and cardiac function. Additionally, it has been reported that EVs (derived from hUCMSCs) also enhanced AMPK activity in rescuing myocardial ischemia [11]. These reports suggest that CBD and EVs combination would be rewarding in improving AMPK since CBD-EVs formulation increased the expression of p-AMPK (Thr 172) in DRG homogenates of PTX-treated mice and DRG neuronal cells. Furthermore, we have analyzed downstream signaling of the AMPK pathway to understand the effects of CBD-EVs against PIPN in mice. 

DRG and spinal homogenates of PTX-treated mice showed lower SirT1 expression, which is consistent with a study by Xiaoning et al., who found that resveratrol treatment alleviated heat and mechanical pain associated with PTX administration in rats by boosting SirT1 expression [79]. AMPK activation elevates intracellular NAD^+^ and stimulates SIRT1, which deacetylates and translocates NRF1 and NRF2 into the nucleus. CBD and CBD-EVs increased NAD^+^ and SirT1 protein expression in DRG and spinal homogenates of PTX-treated animals. These data imply that CBD-EV-regulated AMPK may boost NAD^+^ and SirT1 expression. CBD enhances autophagy and mitochondrial function in human neuroblastoma cells (Parkinsonism in vitro model) by activating the SirT1 pathway. Knockdown of SirT1 by siRNA transfection stops CBD’s autophagy and mitochondrial functional effects against MPP+-induced SHSY5Y cells [80]. In our investigation, we inhibited AMPK activity with compound C (dorsomorphin), and CBD failed to increase SirT1 in PTX-injured DRG primary cells. To our knowledge, this is the first study to show CBD’s impacts on AMPK in neuronal cell mitochondrial health. In keeping with previous results, we also saw NRF1 and NRF2 (GABP) expression in PTX-treated DRG homogenates. NRF1 and NRF2 regulate the transcription of mitochondrial proteins. Enhanced phosphorylation of AMPK activates a plethora of downstream proteins; NRF1/2 regulate mitochondrial biogenesis and function. The transcriptional activity of NRF1/2 regulates TFAM in neuronal cells [81]. PTX reduced NRF1/2 expression in mouse DRG homogenates, whereas CBD restored this effect. In the presence of compound C, CBD failed to increase NRF1/2 expression. In this study, the downregulation of NRF1/2 proteins in peripheral neurons of PTX-treated mice and its upregulation (depending on AMPK activity with CBD and CBD-EVs administration) were first reported.

TFAM has an extensive role in the regulation of the mitochondrial genome and embryonic development in mammalian cells [79]. PTX treatment has been shown to decrease maximal respiration, spare respiratory capacities, ATP production, and mitochondrial membrane depolarisation in isolated DRG neurons [80], which was also observed in the current study. However, in the present study, CBD-EVs treatment showed a more potent significant (*p* < 0.001) effect in reversing mitochondrial membrane depolarization and ATP production in DRG neurons of PTX-treated mice. These results suggest that CBD in EVs formulations has superior mitoprotective effects than CBD alone, and it is important to find out the isolated EVs cargo in detail, including proteins, lipids, nucleic acids, and other cellular components, to understand the mitoprotective role of CBD in EVs formulation. CBD has demonstrated a significant role in mitigating oxidative stress as a direct antioxidant [81]. However, in the present study, we have observed the increased expressions of SOD2, NQO1, and HO1 with CBD and its EVs formulations against PIPN in mice. In addition to regulating mitochondrial bioenergetics, AMPK also activates Nrf2 and NF-κB signaling pathways, which mainly regulate oxidative stress and inflammation [82]. Moreover, phosphorylation of AMPK at the Thr 172 site by therapeutic interventions maintained redox imbalance and inflammatory pool by balancing the Nrf2 and NF-κB pathways [83]. Literature suggests that AMPK activates Nrf2 (human Nrf2) by phosphorylating it at Ser 558, 374, 408, and 433 or by relaying a signal through the p62/autophagy/Keap1 or glycogen synthase kinase 3 (GSK3)/-TrCP-axes, thereby increasing the synthesis of antioxidant enzymes like catalase, hemeoxygenase, superoxide dismutase [84]. In line with these findings, CBD and CBD-EVs treatment in PTX-treated mice increased the expression of antioxidant enzymes and decreased phosphorylation of NF-κB in DRG and spinal tissue homogenates.

In the present study, PTX administration to mice reduced the expressions of 5HT1A and CB1 receptors in DRG homogenates of neuropathic mice, and CBD treatment significantly improved the 5HT1A receptors expression without any effect on CB1 expression. Our findings are supported by a recent study by Sara et al., who reported that CBD offers neuroprotection by selectively activating 5HT1A receptors [19]. Further, 5HT1A receptor agonist treatment protected the retina from oxidative damage and mitochondrial dysfunction [85]. In another study, CBD showed protection against STZ-induced diabetic pain by selectively activating 5HT1A receptors [27]. Pascual et al. demonstrated that administration of CB1/CB2 dual agonist WIN55,212-2 suppressed the thermal hyperalgesia and tactile allodynia induced by PTX in rats, and this effect was blocked by the CB1 antagonist SR141716, suggesting the involvement of the CB1 receptor [86]. In another study, Intraventricular infusion of CBD into the infralimbic cortex of rat brain reduced the anxiety parameter (extinction test) which was hypothesized to be stimulated by activating CB1 receptors [87]. However, Segrado et al., and Sara et al., have shown neither CB1 nor CB2 receptors involvement in CBD’s ability to reduce neuropathic pain [19,88]. Moreover, very limited studies have demonstrated that CBD has less binding affinity for CB1 or CB2 receptors despite its indirect activation of CB1/CB2 receptors by enhancing endocannabinoid levels [89,90]. In agreement with previous reports, we have also observed that CBD anti-nociceptive effects are blocked by the administration of rimonabant, a drug known to block the pharmacological actions of the CB1 receptor, did not improve neurobehavior in PIPN mice (Figure 4). Moreover, it has been demonstrated that 5HT1A receptors play a vital role in the phosphorylation of CaMKII, which is an upstream regulator of the AMPK pathway [21]. Thus, our results may explain the potential of CBD activating 5HT1A receptors and AMPK pathway in regulating mitochondrial bioenergetics, redox and, inflammatory balance of neuronal cells, and activation by CBD may be influencing AMPK pathway in maintaining mitochondrial bioenergetics. However, further studies need to be conducted using knock-out/knock-in animal models to investigate our hypothesis.

## 5. Conclusions

In summary, the primary finding of this study is that CBD-EVs prepared from the sonication method have shown potential in reducing mechanical and thermal pain sensitivities, which are superior to CBD alone against PIPN in mice. CBD and CBD-EVs have potent mitoprotective effects in neuronal cells via activating 5HT1A receptors and the AMPK pathway. CBD was shown to depend on AMPK activation in improving mitochondrial function and biogenesis against PIPN in vitro and in vivo.

## Figures and Tables

**Figure 1 pharmaceutics-15-00554-f001:**
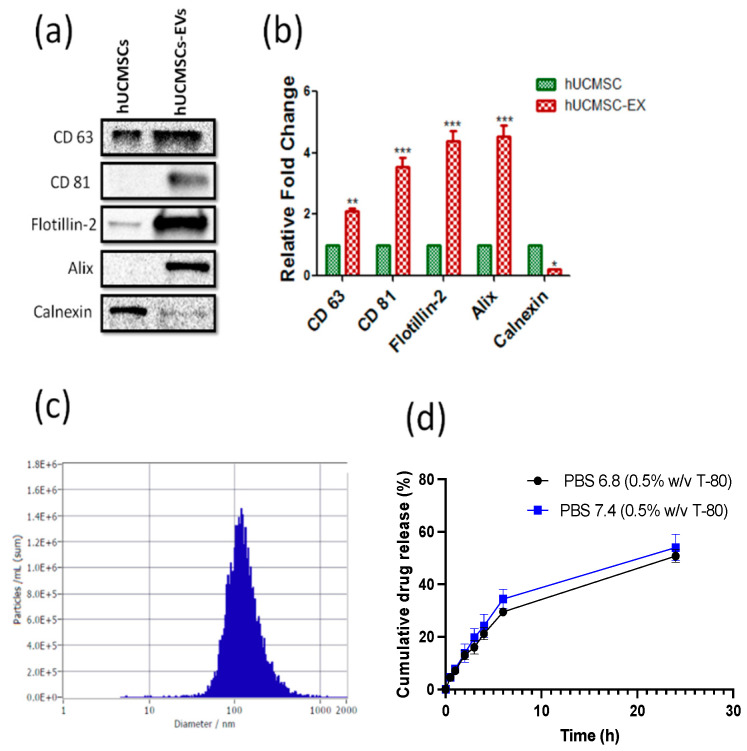
Isolation and characterization of CBD-loaded EVs (hUCMSCs): (**a**). Western blots, and (**b**). densitometric analysis of exosomal markers CD 63, CD 81, Flotillin-2, and stem cell marker; calnexin in human umbilical cord derived stem cells (hUCMSCs) lysates and extracellular vesicles protein lysates derived from hUCMSC (hUCMSCs-EVs). (**c**). Histogram represents the mean particle size distribution of CBD-EVs and (**d**). representative line plot showing % cumulative release of CBD from extracellular vesicles in PBS at pH 7.4 and 6.8, respectively, at different time points. (Statistical significance: *; *p* < 0.05, **; *p* < 0.01 and ***; *p* < 0.001).

**Figure 2 pharmaceutics-15-00554-f002:**
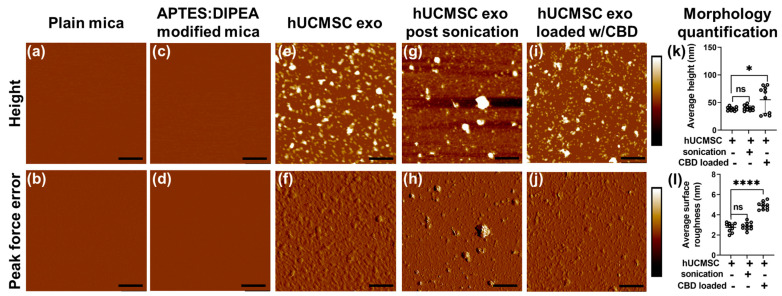
Morphology and corresponding quantification of human mesenchymal stem cell (hUCMSC) EVs under various conditions. Representative images of plain mica (**a**) Height image (**b**) Peak force error image. Representative images of APTES: DIPEA modified mica (**c**) Height image (**d**) Peak force error image. Representative images of hUCMSC-derived EVs (**e**) Height image (**f**) Peak force error image. Representative images of hUCMSC EVs post sonication (control) (**g**) Height image (**h**) Peak force error image. Representative images of hUCMSC EVs loaded with CBD (**i**) Height image (**j**) Peak force error image. (Scale bar: 1 µm; color bar for height image: −65 nm to 65 nm; color bar for peak force error image: −300 pN to 300 pN). Morphology quantification (**k**) Average height (**l**) Average surface roughness. (Statistical significance: ns; not significant. *, *p* < 0.05 and **** *p* < 0.0001).

**Figure 3 pharmaceutics-15-00554-f003:**
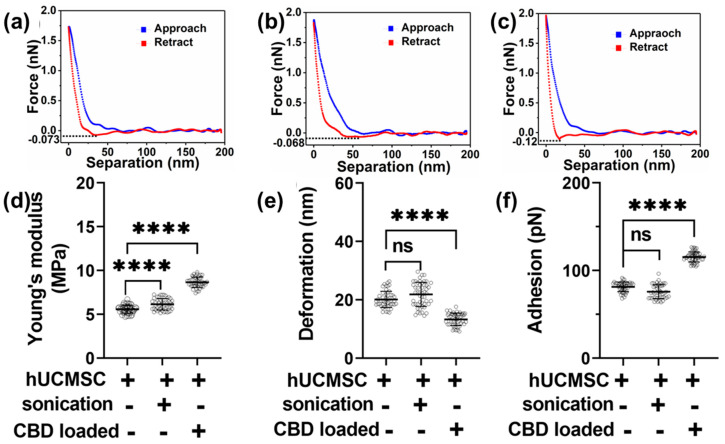
Nanomechanical characteristics of human mesenchymal stem cell (hUCMSC) EVs under various conditions. A representative force-separation curve displaying adhesion value corresponding to EVs derived from (**a**) hUCMSC, (**b**) hUCMSC post sonication (control), (**c**) CBD loaded hUCMSC. Nanomechanical attributes displaying (**d**) Young’s modulus, (**e**) Deformation, and (**f**) Adhesion. (Statistical significance: ns; not significant. ****; *p* < 0.0001).

**Figure 4 pharmaceutics-15-00554-f004:**
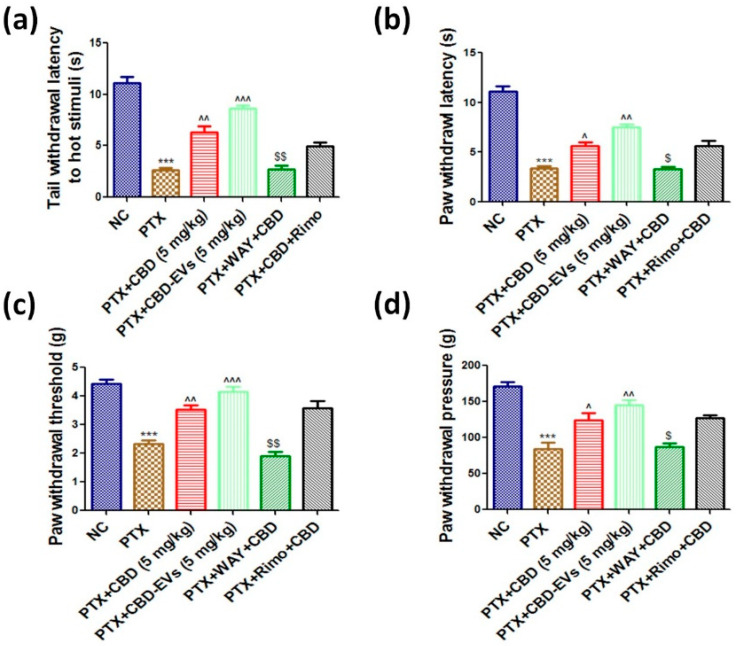
Effect of CBD and CBD-EVS on neurobehavior of PTX-induced neuropathic mice. Bar graphs represent (**a**) Hot immersion test (**b**) Hargreaves plantar test, (**c**) Vonfrey test, and (**d**) Randall selitto test; values are expressed as mean ± SEM (n = 3). *** *p* < 0.001 vs. NC, ^ *p* < 0.05, ^^ *p* < 0.01 and ^^^ *p* < 0.001 vs. PTX (8 mg/kg), $ *p* < 0.05 and $$ *p* < 0.01 vs. PTX + CBD (5 mg/kg). NC: untreated age-matched mice, PTX: mice received (8 mg/kg/day) on alternate days for four injections cumulatively, PTX + CBD (5 mg/kg): 5 mg/kg CBD (i.p.) injected twice a week for six weeks after the last PTX injection, PTX + CBD-EVs: 5 mg/kg of CBD loaded in hUCMSCs-derived EVs (i.p.) injected twice a week for six weeks after the last dose of PTX injection, PTX + WAY + CBD: 10 mg/kg/day, i.p., WAY100135 (5HT1A receptor inhibitor) was given to mice for four weeks and three hours before administering CBD, PTX + CBD + RIMO: 3 mg/kg/day, i.p., rimonabant (CB1 receptor blocker) was given to mice for four weeks and three hours before administering CBD.

**Figure 5 pharmaceutics-15-00554-f005:**
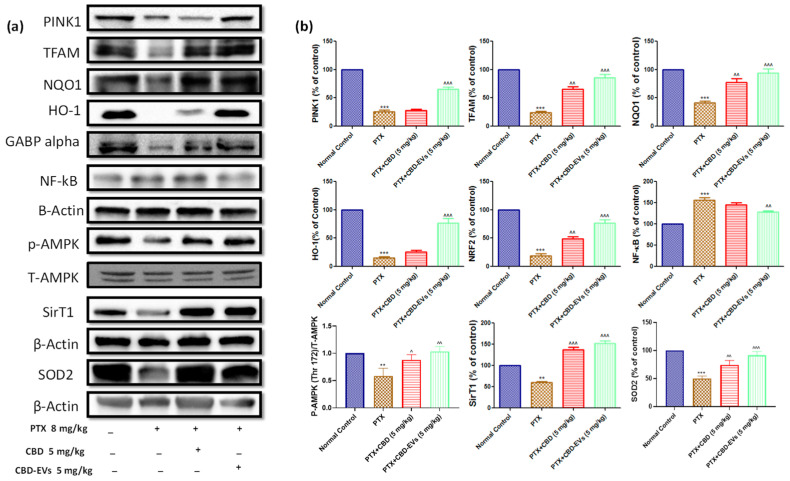
Effect of CBD and CBD-EVs on AMPK-SIRT1-NRF1/2 Axis. (**a**) Western blots of DRG homogenates from PTX-treated mice show treatment with CBD (5 mg/kg) and CBD-EVs (5 mg/kg) for six weeks after last dose of PTX administration. (**b**) Bar graphs represent the respective western blots quantification. Values are expressed as mean ± SEM (n = 3). ** *p* < 0.01 and *** *p* < 0.001 vs. Normal control, ^ *p* < 0.05, ^^ *p* < 0.01 and ^^^ *p* < 0.001 vs. PTX (8 mg/kg). NC: untreated age-matched mice, PTX: mice received (8 mg/kg/day) on alternate days for four injections cumulatively, PTX + CBD (5 mg/kg): 5 mg/kg CBD (i.p.) injected twice a week for six weeks after the last PTX injection, PTX + CBD − EVs: 5 mg/kg of CBD loaded in hUCMSCs-derived EVs (i.p.) injected twice a week for six weeks after the last dose of PTX injection.

**Figure 6 pharmaceutics-15-00554-f006:**
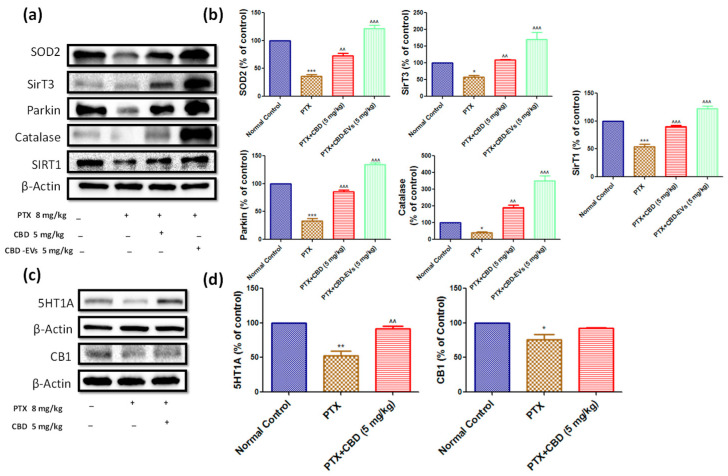
Effect of CBD and CBDEVs on AMPK-SIRT1-NRF1/2 Axis in spinal homogenates and effect of CBD on CB1 and 5HT1A receptors expression in DRG homogenates of PTX induced neuropathic mice. (**a**) Western blots of spinal homogenates show treatment with CBD (5 mg/kg) and CBD-EVs (5 mg/kg) and (**c**) Western blots of DRG homogenates show treatment with CBD (5 mg/kg) in PTX-treated mice for six weeks after last dose of PTX administration. (**b**,**d**) Bar graphs represent the respective western blots quantification. Values are expressed as mean ± SEM (n = 3). * *p* < 0.05, ** *p* < 0.01 and *** *p* < 0.001 vs. Normal control, ^^ *p* < 0.01 and ^^^ *p* < 0.001 vs. PTX (8 mg/kg). NC: untreated age-matched mice, PTX: mice received (8 mg/kg/day) on alternate days for four injections cumulatively, PTX + CBD (5 mg/kg): 5 mg/kg CBD (i.p.) injected twice a week for six weeks after the last PTX injection, PTX + CBD-EVs: 5 mg/kg of CBD loaded in hUCMSCs-derived EVs (i.p.) injected twice a week for six weeks after the last dose of PTX injection.

**Figure 7 pharmaceutics-15-00554-f007:**
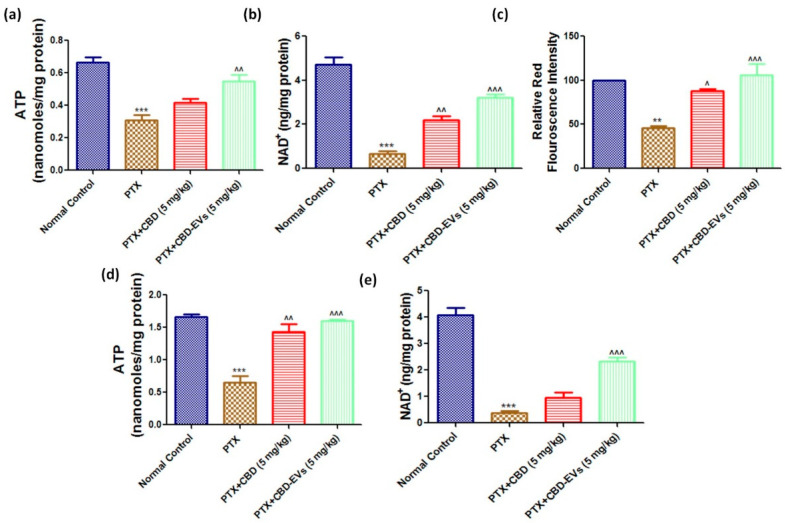
Effect of CBD and CBD-EVs on Mitochondrial Function and mitochondrial membrane potential (ΔΨm). (**a**) Representative bar graphs showing the ATP levels in Fresh DRG homogenates and (**d**) spinal homogenates of PTX-treated mice, (**b**) bar graph showing NAD^+^ levels in DRG homogenates and (**e**) in spinal homogenates of PTX-treated mice, (**c**) Bar graph represents the JC1 trimer aggregates relative red fluorescence intensity. Values are expressed as mean ± SEM (n = 3). ** *p* < 0.01 and *** *p* < 0.001 vs. Normal control, ^ *p* < 0.05, ^^ *p* < 0.01 and ^^^ *p* < 0.001 vs. PTX (8 mg/kg). NC: untreated age-matched mice, PTX: mice received (8 mg/kg/day) on alternate days for four injections cumulatively, PTX + CBD (5 mg/kg): 5 mg/kg CBD (i.p.) injected twice a week for six weeks after the last PTX injection, PTX + CBD-EVs: 5 mg/kg of CBD loaded in hUCMSCs-derived EVs (i.p.) injected twice a week for six weeks after the last dose of PTX injection.

**Figure 8 pharmaceutics-15-00554-f008:**
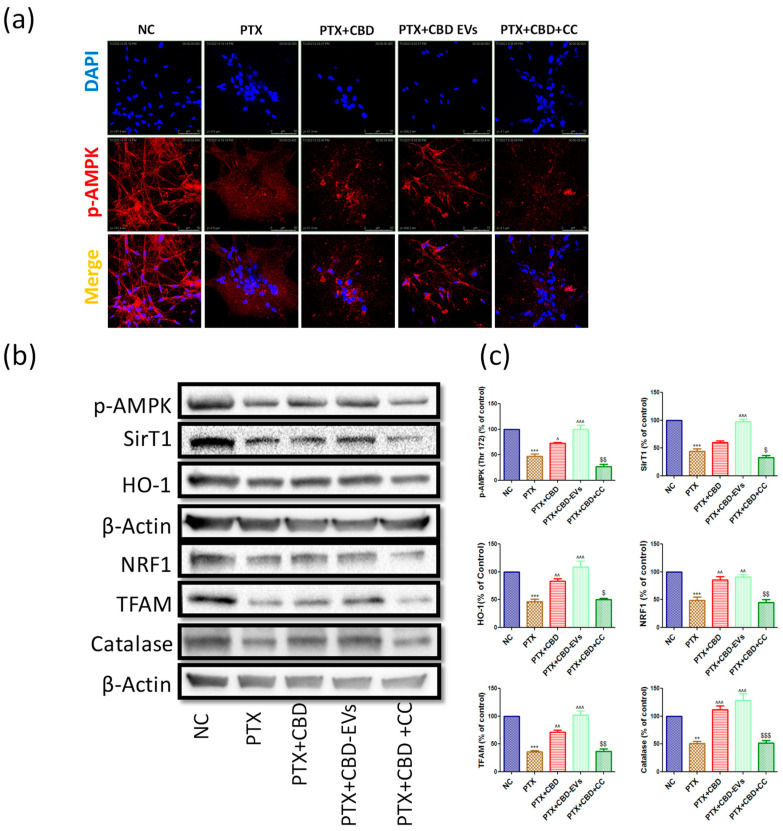
Immunoexpressions of the AMPK pathway in cultured DRG primary cells (**a**) Representative confocal microscope images showing the upper panel nuclear (DAPI) staining, middle panel: immuno expression of p-AMPK (Thr 172) labeled with rhodamine, and lower panel showing the merge of p-AMPK and DAPI (**b**) Western blots of cultured DRG lysates show treatment with 12 μM either of CBD and CBD-EVs in PTX-treated neuronal cells for 48 h (**c**) Bar graphs represent the respective western blots quantification. ** *p* < 0.01 and *** *p* < 0.001 vs. NC, ^ *p* < 0.05, ^^ *p* < 0.01 and ^^^ *p* < 0.001 vs. PTX (8 mg/kg). $ *p* < 0.05, $$ *p* < 0.01 and $$$ *p* < 0.001 vs. PTX + CBD. NC: Untreated adult rat DRG primary cells, PTX: The primary cells (DRG cultures) were grown and treated with 3 µM paclitaxel for 48 h, PTX + CBD: The primary cells (DRG cultures) were grown and treated with 3 µM paclitaxel followed by post-treatment with 12 µM of CBD for 48 h, PTX + CBD-EVs: The primary cells (DRG cultures) were grown and treated with 3 µM paclitaxel followed by post-treatment with 12 µM of CBD-EVs for 48 h, PTX + CBD + CC: The primary cells (DRG cultures) were grown and treated with 3 µM paclitaxel followed by post-treatment with 12 µM of CBD and 10 µM of compound C for 48 h.

**Table 1 pharmaceutics-15-00554-t001:** Effect of EVs, CBD, and CBD-EVs on PTX-induced neurobehavioral changes: The values are expressed as mean standard error of the mean (n = 6). Normal Control: untreated age-matched mice, PTX: mice received (8 mg/kg/day) on alternate days for four injections cumulatively, PTX + EVs: Mice were given extracellular vesicles protein (10 mg/kg, i.p.) twice a week for six weeks after the last dose of PTX. PTX + CBD (5 mg/kg): 5 mg/kg CBD (i.p.) injected twice a week for six weeks after the last PTX injection, PTX + CBD + EVs: 5 mg/kg of CBD and hUCMSCs-derived EVs (i.p.) injected twice a week for six weeks after the last dose of PTX injection. PTX + CBD-EVs: 5 mg/kg of CBD loaded in hUCMSCs-derived EVs (i.p.) injected twice a week for six weeks after the last dose of PTX injection. *** *p* < 0.001 vs. Normal control and ^ *p* < 0.05, ^^ *p* < 0.01 and ^^^ *p* < 0.001 vs. PTX.

Parameter	Normal Control	PTX	PTX + EVs	PTX + CBD (5 mg/kg)	PTX + EVs + CBD	PTX + CBD-EVs (5 mg/kg)
Paw withdrawal latency to hot stimuli (s)	10.86 ± 0.36	4.05 ± 0.31 ***	6.41 ± 0.98 ^	7.3 ± 0.4 ^^	6.93 ± 0.82 ^^	9.8 ± 0.66 ^^^
Paw withdrawal latency to cold stimuli (s)	13.43 ± 0.43	5.78 ± 0.24 ***	7.9 ± 0.61 ^	8.1 ± 0.31 ^^	8.23 ± 0.93 ^^	10.09 ± 0.8 ^^^
Paw withdrawal latency (s)	10.89 ± 0.40	3.85 ± 0.8 ***	5.99 ± 0.91 ^	6.86 ± 0.9 ^	6.79 ± 0.79 ^	8.10 ± 0.45 ^^^
Paw withdrawal threshold (g)	4.63 ± 0.24	1.91 ± 0.36 ***	2.97 ± 0.6 ^	3.13 ± 0.1 ^	3.11 ± 0.87 ^	4.19 ± 0.7 ^^^
Average Body weight (g)	23.18 ± 1.11	22.65 ± 1.23	23.85 ± 1.99	23.99 ± 1.20	23.03 ± 1.08	23.21 ± 1.32

## Data Availability

The data that support the findings of this study are available on request from the corresponding author.

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
