# Peer review of "Cannabidiol-Loaded Extracellular Vesicles from Human Umbilical Cord Mesenchymal Stem Cells Alleviate Paclitaxel-Induced Peripheral Neuropathy"

_pharmaceutics, 2023, doi:10.3390/pharmaceutics15020554_

Round 1

Reviewer 1 Report

The manuscript is very interest it well structure and with a good use of statistical analysis.

The tables and figures are clear.

The conclusion is clearly and concise

In general is a good manuscript.

Author Response

Thank you for your comments. 

Reviewer 2 Report

The study by Kalvala et al. has the interesting purpose of exploring the potential of CBD to support effective chemotherapy. The manuscript needs urgent major revisions in many parts which compromise understanding of the study.

Here are the comments:

line 28: paraphrase

lines 28-29: paraphrase

lines 39-41: how long the treatment with this PTX usually takes?

line 42: explain the “nab-ptx” abbreviation

line 44: what population? 

lines 52-55: this paragraph is out of place as an interruption of the introductory flow, it has a beginning and an end not in support of the aim of the study.

line 66: specify the abbreviation at its first appearance in the text. 

line 75: correct punctuation

lines 70-73: EVs use deduction may not follow an explanation of AFM use. It should be added after the description of the rationale for the use of EVs 

79-80: the sentence about cannabinoids is not clear at all. The authors should describe in more detail the AMPK pathway and the activation by hUCMSCS.

line 94: this CBD paragraph should be merged with the previous one

lines 94-100: what does the mitochondrium have to do with it? the previous explanation, which the authors should enrich more, should be moved to this paragraph.

Authors should also specify what the markers of the PIPN are before describing the aim.

103-104: the authors should specify the “cannabinoid- and non-cannabinoid receptors” target and why. A more detailed description of the pluri-target mechanism of CBD must be added here.

the AFM employment in the study is not described in the final part of the intro, as the aim of the study. 

How has the dosage of CBD been chosen?

Why did the authors choose to carry out the study on mice and rats? What about the age?

The authors should explain the rationale for the administration of CBD-EVS after treatment with PTZ and not during.

The authors should not repeat the abbreviations if already specified.

line 213: how did the authors “confirm neuropathy in mice”?

The description of the groups is confusing and the control group for the CBD is missing. Above all, the authors should specify why they did not carry out the CBD control study, i.e. administration of CBD without PTX.

The authors should add statistical details besides the p value.

The authors should mention RIMONABANT and WAY before the result section. There is no mention in materials or animals sub-sections. 

The discussion is disorganized and this does not allow highlighting the results.

Author Response

Dear Reviewer,

The attached file has point-by-point responses to the reviewer’s comments repeated in italics, followed by our responses.

Thank you. 

Reviewer 3 Report

The Authors can find the revisions in the attached file.

Author Response

The attached file has point-by-point responses to the reviewer’s comments repeated in italics, followed by our responses.

Thank you. 

Round 2

Reviewer 2 Report

The Authors replied properly to the comments. All the modifications and clarifications required were presented. 

Reviewer 3 Report

The Authors have addressed all the required revision and the manuscript in the present form is well described and discussed, fluent and comprehensible. Overall the manuscript has greatly improved, therfore I reccommend accepting the manuscript for publication.